# Biologically Inspired Dynamic Textures for Probing Motion Perception

**Jonathan Vacher**
CNRS UNIC and Ceremade
Univ. Paris-Dauphine
75775 Paris Cedex 16, FRANCE
vacher@ceremade.dauphine.fr

**Andrew Isaac Meso**
Institut de Neurosciences de la Timone
UMR 7289 CNRS/Aix-Marseille Université
13385 Marseille Cedex 05, FRANCE
andrew.meso@univ-amu.fr

**Laurent Perrinet**
Institut de Neurosciences de la Timone
UMR 7289 CNRS/Aix-Marseille Université
13385 Marseille Cedex 05, FRANCE
laurent.perrinet@univ-amu.fr

**Gabriel Peyré**
CNRS and Ceremade
Univ. Paris-Dauphine
75775 Paris Cedex 16, FRANCE
peyre@ceremade.dauphine.fr

## Abstract

Perception is often described as a predictive process based on an optimal inference with respect to a generative model. We study here the principled construction of a generative model specifically crafted to probe motion perception. In that context, we first provide an axiomatic, biologically-driven derivation of the model. This model synthesizes random dynamic textures which are defined by stationary Gaussian distributions obtained by the random aggregation of warped patterns. Importantly, we show that this model can equivalently be described as a stochastic partial differential equation. Using this characterization of motion in images, it allows us to recast motion-energy models into a principled Bayesian inference framework. Finally, we apply these textures in order to psychophysically probe speed perception in humans. In this framework, while the likelihood is derived from the generative model, the prior is estimated from the observed results and accounts for the perceptual bias in a principled fashion.

## 1 Motivation

A normative explanation for the function of perception is to infer relevant hidden parameters from the sensory input with respect to a generative model [7]. Equipped with some prior knowledge about this representation, this corresponds to the *Bayesian brain* hypothesis, as has been perfectly illustrated by the particular case of motion perception [19]. However, the Gaussian hypothesis related to the parameterization of knowledge in these models —for instance in the formalization of the prior and of the likelihood functions— does not always fit with psychophysical results [17]. As such, a major challenge is to refine the definition of generative models so that they conform to the widest variety of results.

From this observation, the estimation problem inherent to perception is linked to the definition of an adequate generative model. In particular, the simplest generative model to describe visual motion is the luminance conservation equation. It states that luminance $I(x,t)$ for $(x,t) \in \mathbb{R}^2 \times \mathbb{R}$ is approximately conserved along trajectories defined as integral lines of a vector field $v(x,t) \in \mathbb{R}^2 \times \mathbb{R}$. The corresponding generative model defines random fields as solutions to the stochastic partial differential equation (sPDE),

$$\langle v, \nabla I \rangle + \frac{\partial I}{\partial t} = W, \tag{1}$$

where $\langle \cdot, \cdot \rangle$ denotes the Euclidean scalar product in $\mathbb{R}^2$, $\nabla I$ is the spatial gradient of $I$. To match the statistics of natural scenes or some category of textures, the driving term $W$ is usually defined as a colored noise corresponding to some average spatio-temporal coupling, and is parameterized by a covariance matrix $\Sigma$, while the field is usually a constant vector $v(x,t) = v_0$ accounting for a full-field translation with constant speed.

Ultimately, the application of this generative model is essential for probing the visual system, for instance to understand how observers might detect motion in a scene. Indeed, as shown by [9, 19], the negative log-likelihood corresponding to the luminance conservation model (1) and determined by a hypothesized speed $v_0$ is proportional to the value of the motion-energy model [1] $\|\langle v_0, \nabla(K \star I) \rangle + \frac{\partial (K \star I)}{\partial t}\|^2$, where $K$ is the whitening filter corresponding to the inverse of $\Sigma$, and $\star$ is the convolution operator. Using some prior knowledge on the distribution of motions, for instance a preference for slow speeds, this indeed leads to a Bayesian formalization of this inference problem [18]. This has been successful in accounting for a large class of psychophysical observations [19]. As a consequence, such probabilistic frameworks allow one to connect different models from computer vision to neuroscience with a unified, principled approach.

However the model defined in (1) is obviously quite simplistic with respect to the complexity of natural scenes. It is therefore useful here to relate this problem to solutions proposed by texture synthesis methods in the computer vision community. Indeed, the literature on the subject of static textures synthesis is abundant (see [16] and the references therein for applications in computer graphics). Of particular interest for us is the work of Galerne et al. [6], which proposes a stationary Gaussian model restricted to static textures. Realistic dynamic texture models are however less studied, and the most prominent method is the non-parametric Gaussian auto-regressive (AR) framework of [3], which has been refined in [20].

**Contributions.**   Here, we seek to engender a better understanding of motion perception by improving generative models for dynamic texture synthesis. From that perspective, we motivate the generation of optimal stimulation within a stationary Gaussian dynamic texture model. We base our model on a previously defined heuristic [10, 11] coined "Motion Clouds". Our first contribution is

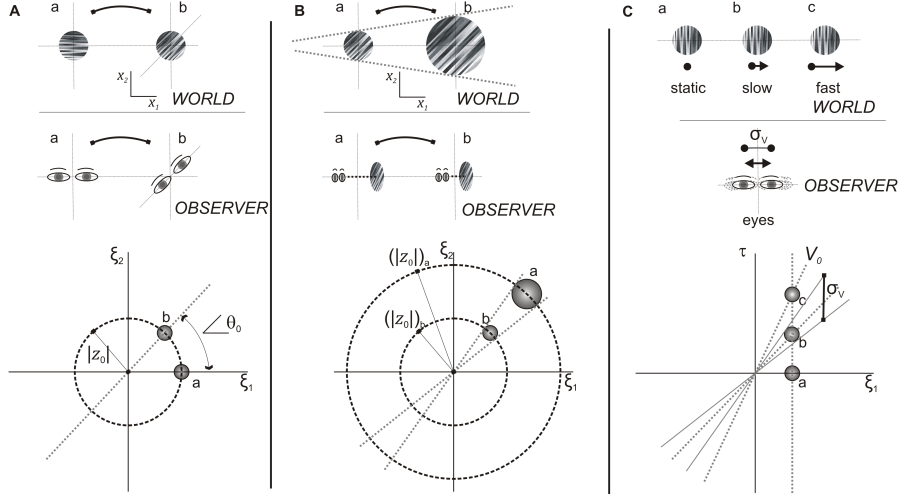

**Figure 1:** *Parameterization of the class of Motion Clouds stimuli.* The illustration relates the parametric changes in MC with real world (top row) and observer (second row) movements. **(A)** Orientation changes resulting in scene rotation are parameterized through $\theta$ as shown in the bottom row where a horizontal $a$ and obliquely oriented $b$ MC are compared. **(B)** Zoom movements, either from scene looming or observer movements in depth, are characterised by scale changes reflected by a scale or frequency term $z$ shown for a larger or closer object $b$ compared to more distant $a$. **(C)** Translational movements in the scene characterised by $V$ using the same formulation for static (a) slow (b) and fast moving MC, with the variability in these speeds quantified by $\sigma_V$. ($\xi$ and $\tau$) in the third row are the spatial and temporal frequency scale parameters. The development of this formulation is detailed in the text.

an axiomatic derivation of this model, seen as a shot noise aggregation of dynamically warped "textons". This formulation is important to provide a clear understanding of the effects of the model's parameters manipulated during psychophysical experiments. Within our generative model, they correspond to average translation speed and orientation of the "textons" and standard deviations of random fluctuations around this average. Our second contribution (proved in the supplementary materials) is to demonstrate an explicit equivalence between this model and a class of linear stochastic partial differential equations (sPDE). This shows that our model is a generalization of the well-known luminance conservation equation. This sPDE formulation has two chief advantages: it allows for a real-time synthesis using an AR recurrence and it allows one to recast the log-likelihood of the model as a generalization of the classical motion energy model, which in turn is crucial to allow for a Bayesian modeling of perceptual biases. Our last contribution is an illustrative application of this model to the psychophysical study of motion perception in humans. This application shows how the model allows us to define a likelihood, which enables a simple fitting procedure to determine the prior driving the perceptual bias.

**Notations.** In the following, we will denote $(x, t) \in \mathbb{R}^2 \times \mathbb{R}$ the space/time variable, and $(\xi, \tau) \in \mathbb{R}^2 \times \mathbb{R}$ the corresponding frequency variables. If $f(x, t)$ is a function defined on $\mathbb{R}^3$, then $\hat{f}(\xi, \tau)$ denotes its Fourier transform. For $\xi \in \mathbb{R}^2$, we denote $\xi = \|\xi\|(\cos(\angle\xi), \sin(\angle\xi)) \in \mathbb{R}^2$ its polar coordinates. For a function $g$ in $\mathbb{R}^2$, we denote $\bar{g}(x) = g(-x)$. In the following, we denote with a capital letter such as $A$ a random variable, a we denote $a$ a realization of $A$, we let $\mathbb{P}_A(a)$ be the corresponding distribution of $A$.

## 2 Axiomatic Construction of a Dynamic Texture Stimulation Model

Solving a model-based estimation problem and finding optimal dynamic textures for stimulating an instance of such a model can be seen as equivalent mathematical problems. In the luminance conservation model (1), the generative model is parameterized by a spatio-temporal coupling function, which is encoded in the covariance $\Sigma$ of the driving noise and the motion flow $v_0$. This coupling (covariance) is essential as it quantifies the extent of the spatial integration area as well as the integration dynamics, an important issue in neuroscience when considering the implementation of integration mechanisms from the local to the global scale. In particular, it is important to understand modular sensitivity in the various lower visual areas with different spatio-temporal selectivities such as Primary Visual Cortex (V1) or ascending the processing hierarchy, Middle Temple area (MT). For instance, by varying the frequency bandwidth of such dynamic textures, distinct mechanisms for perception and action have been identified [11]. However, such textures were based on a heuristic [10], and our goal here is to develop a principled, axiomatic definition.

### 2.1 From Shot Noise to Motion Clouds

We propose a mathematically-sound derivation of a general parametric model of dynamic textures. This model is defined by aggregation, through summation, of a basic spatial "texton" template $g(x)$. The summation reflects a transparency hypothesis, which has been adopted for instance in [6]. While one could argue that this hypothesis is overly simplistic and does not model occlusions or edges, it leads to a tractable framework of stationary Gaussian textures, which has proved useful to model static micro-textures [6] and dynamic natural phenomena [20]. The simplicity of this framework allows for a fine tuning of frequency-based (Fourier) parameterization, which is desirable for the interpretation of psychophysical experiments.

We define a random field as

$$I_\lambda(x, t) \stackrel{\text{def.}}{=} \frac{1}{\sqrt{\lambda}} \sum_{p \in \mathbb{N}} g(\varphi_{A_p}(x - X_p - V_p t)) \tag{2}$$

where $\varphi_a : \mathbb{R}^2 \to \mathbb{R}^2$ is a planar warping parameterized by a finite dimensional vector $a$. Intuitively, this model corresponds to a dense mixing of stereotyped, static textons as in [6]. The originality is two-fold. First, the components of this mixing are derived from the texton by visual transformations $\varphi_{A_p}$ which may correspond to arbitrary transformations such as zooms or rotations, illustrated in Figure 1. Second, we explicitly model the motion (position $X_p$ and speed $V_p$) of each individual texton. The parameters $(X_p, V_p, A_p)_{p \in \mathbb{N}}$ are independent random vectors. They account for the

variability in the position of objects or observers and their speed, thus mimicking natural motions in an ambient scene. The set of translations $(X_p)_{p \in \mathbb{N}}$ is a 2-D Poisson point process of intensity $\lambda > 0$. The following section instantiates this idea and proposes canonical choices for these variabilities. The warping parameters $(A_p)_p$ are distributed according to a distribution $\mathbb{P}_A$. The speed parameters $(V_p)_p$ are distributed according to a distribution $\mathbb{P}_V$ on $\mathbb{R}^2$. The following result shows that the model (2) converges to a stationary Gaussian field and gives the parameterization of the covariance. Its proof follows from a specialization of [5, Theorem 3.1] to our setting.

**Proposition 1.** $I_\lambda$ *is stationary with bounded second order moments. Its covariance is* $\Sigma(x, t, x', t') = \gamma(x - x', t - t')$ *where $\gamma$ satisfies*

$$\forall (x, t) \in \mathbb{R}^3, \quad \gamma(x, t) = \int \int_{\mathbb{R}^2} c_g(\varphi_a(x - \nu t)) \mathbb{P}_V(\nu) \mathbb{P}_A(a) \mathrm{d}\nu \mathrm{d}a \tag{3}$$

*where $c_g = g \star \bar{g}$ is the auto-correlation of $g$. When $\lambda \to +\infty$, it converges (in the sense of finite dimensional distributions) toward a stationary Gaussian field $I$ of zero mean and covariance $\Sigma$.*

## 2.2 Definition of "Motion Clouds"

We detail this model here with warpings as rotations and scalings (see Figure 1). These account for the characteristic orientations and sizes (or spatial scales) in a scene with respect to the observer

$$\forall a = (\theta, z) \in [-\pi, \pi) \times \mathbb{R}_+^*, \quad \varphi_a(x) \overset{\text{def.}}{=} z R_{-\theta}(x),$$

where $R_\theta$ is the planar rotation of angle $\theta$. We now give some physical and biological motivation underlying our particular choice for the distributions of the parameters. We assume that the distributions $\mathbb{P}_Z$ and $\mathbb{P}_\Theta$ of spatial scales $z$ and orientations $\theta$, respectively (see Figure 1), are independent and have densities, thus considering $\forall a = (\theta, z) \in [-\pi, \pi) \times \mathbb{R}_+^*, \quad \mathbb{P}_A(a) = \mathbb{P}_Z(z) \mathbb{P}_\Theta(\theta)$. The speed vector $\nu$ is assumed to be randomly fluctuating around a central speed $v_0$, so that

$$\forall \nu \in \mathbb{R}^2, \quad \mathbb{P}_V(\nu) = \mathbb{P}_{\|V - v_0\|}(\|\nu - v_0\|). \tag{4}$$

In order to obtain "optimal" responses to the stimulation (as advocated by [21]), it makes sense to define the texton $g$ to be equal to an oriented Gabor acting as an atom, based on the structure of a standard receptive field of V1. Each would have a scale $\sigma$ and a central frequency $\xi_0$. Since the orientation and scale of the texton is handled by the $(\theta, z)$ parameters, we can impose without loss of generality the normalization $\xi_0 = (1, 0)$. In the special case where $\sigma \to 0$, $g$ is a grating of frequency $\xi_0$, and the image $I$ is a dense mixture of drifting gratings, whose power-spectrum has a closed form expression detailed in Proposition 2. Its proof can be found in the supplementary materials. We call this Gaussian field a Motion Cloud (MC), and it is parameterized by the envelopes $(\mathbb{P}_Z, \mathbb{P}_\Theta, \mathbb{P}_V)$ and has central frequency and speed $(\xi_0, v_0)$. Note that it is possible to consider any arbitrary textons $g$, which would give rise to more complicated parameterizations for the power spectrum $\hat{g}$, but we decided here to stick to the simple case of gratings.

**Proposition 2.** *When $g(x) = e^{\mathrm{i}\langle x, \xi_0 \rangle}$, the image $I$ defined in Proposition 1 is a stationary Gaussian field of covariance having the power-spectrum*

$$\forall (\xi, \tau) \in \mathbb{R}^2 \times \mathbb{R}, \ \hat{\gamma}(\xi, \tau) = \frac{\mathbb{P}_Z(\|\xi\|)}{\|\xi\|^2} \mathbb{P}_\Theta(\angle \xi) \mathcal{L}(\mathbb{P}_{\|V - v_0\|}) \left( -\frac{\tau + \langle v_0, \xi \rangle}{\|\xi\|} \right), \tag{5}$$

*where the linear transform $\mathcal{L}$ is such that $\forall u \in \mathbb{R}, \mathcal{L}(f)(u) = \int_{-\pi}^{\pi} f(-u/\cos(\varphi)) \mathrm{d}\varphi$.*

*Remark* 1. Note that the envelope of $\hat{\gamma}$ is shaped along a cone in the spatial and temporal domains. This is an important and novel contribution when compared to a Gaussian formulation like a classical Gabor. In particular, the bandwidth is then constant around the speed plane or the orientation line with respect to spatial frequency. Basing the generation of the textures on all possible translations, rotations and zooms, we thus provide a principled approach to show that bandwidth should be proportional to spatial frequency to provide a better model of moving textures.

## 2.3 Biologically-inspired Parameter Distributions

We now give meaningful specialization for the probability distributions $(\mathbb{P}_Z, \mathbb{P}_\Theta, \mathbb{P}_{\|V - v_0\|})$, which are inspired by some known scaling properties of the visual transformations relevant to dynamic scene perception.

First, small, centered, linear movements of the observer along the axis of view (orthogonal to the plane of the scene) generate centered planar zooms of the image. From the linear modeling of the observer's displacement and the subsequent multiplicative nature of zoom, scaling should follow a Weber-Fechner law stating that subjective sensation when quantified is proportional to the logarithm of stimulus intensity. Thus, we choose the scaling $z$ drawn from a log-normal distribution $\mathbb{P}_Z$, defined in (6). The bandwidth $\sigma_Z$ quantifies the variance in the amplitude of zooms of individual textons relative to the set characteristic scale $z_0$. Similarly, the texture is perturbed by variation in the global angle $\theta$ of the scene: for instance, the head of the observer may roll slightly around its normal position. The von-Mises distribution – as a good approximation of the warped Gaussian distribution around the unit circle – is an adapted choice for the distribution of $\theta$ with mean $\theta_0$ and bandwidth $\sigma_\Theta$, see (6). We may similarly consider that the position of the observer is variable in time. On first order, movements perpendicular to the axis of view dominate, generating random perturbations to the global translation $v_0$ of the image at speed $\nu - v_0 \in \mathbb{R}^2$. These perturbations are for instance described by a Gaussian random walk: take for instance tremors, which are constantly jittering, small ($\leqslant 1$ deg) movements of the eye. This justifies the choice of a radial distribution (4) for $\mathbb{P}_V$. This radial distribution $\mathbb{P}_{\|V-v_0\|}$ is thus selected as a bell-shaped function of width $\sigma_V$, and we choose here a Gaussian function for simplicity, see (6). Note that, as detailed in the supplementary a slightly different bell-function (with a more complicated expression) should be used to obtain an exact equivalence with the sPDE discretization mentioned in Section 4.

The distributions of the parameters are thus chosen as

$$\mathbb{P}_Z(z) \propto \frac{z_0}{z} e^{-\frac{\ln\left(\frac{z}{z_0}\right)^2}{2\ln\left(1+\sigma_Z^2\right)}}, \quad \mathbb{P}_\Theta(\theta) \propto e^{\frac{\cos(2(\theta-\theta_0))}{4\sigma_\Theta^2}} \quad \text{and} \quad \mathbb{P}_{\|V-v_0\|}(r) \propto e^{-\frac{r^2}{2\sigma_V^2}}. \tag{6}$$

*Remark* 2. Note that in practice we have parametrized $\mathbb{P}_Z$ by its mode $m_Z = \mathrm{argmax}_z\, \mathbb{P}_Z(z)$ and standard deviation $d_Z = \sqrt{\int z^2 \mathbb{P}_Z(z)\mathrm{d}z}$, see the supplementary material and [4].

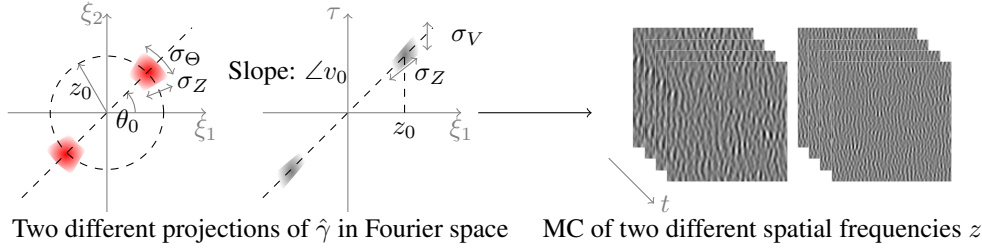

Two different projections of $\hat{\gamma}$ in Fourier space    MC of two different spatial frequencies $z$

**Figure 2:** Graphical representation of the covariance $\gamma$ (left) —note the cone-like shape of the envelopes– and an example of synthesized dynamics for narrow-band and broad-band Motion Clouds (right).

Plugging these expressions (6) into the definition (5) of the power spectrum of the motion cloud, one obtains a parameterization which is very similar to the one originally introduced in [11]. The following table gives the speed $v_0$ and frequency $(\theta_0, z_0)$ central parameters in terms of amplitude and orientation, each one being coupled with the relevant dispersion parameters. Figure 1 and 2 shows a graphical display of the influence of these parameters.

|  | Speed | Freq. orient. | Freq. amplitude |
|---|---|---|---|
| (mean, dispersion) | $(v_0, \sigma_V)$ | $(\theta_0, \sigma_\Theta)$ | $(z_0, \sigma_Z)$ or $(m_Z, d_Z)$ |

*Remark* 3. Note that the final envelope of $\hat{\gamma}$ is in agreement with the formulation that is used in [10]. However, that previous derivation was based on a heuristic which intuitively emerged from a long interaction between modelers and psychophysicists. Herein, we justified these different points from first principles.

*Remark* 4. The MC model can equally be described as a stationary solution of a stochastic partial differential equation (sPDE). This sPDE formulation is important since we aim to deal with dynamic stimulation, which should be described by a causal equation which is local in time. This is crucial for numerical simulations, since, this allows us to perform real-time synthesis of stimuli using an

auto-regressive time discretization. This is a significant departure from previous Fourier-based implementation of dynamic stimulation [10, 11]. This is also important to simplify the application of MC inside a bayesian model of psychophysical experiments (see Section 3)The derivation of an equivalent sPDE model exploits a spectral formulation of MCs as Gaussian Random fields. The full proof along with the synthesis algorithm can be found in the supplementary material.

# 3 Psychophysical Study: Speed Discrimination

To exploit the useful features of our MC model and provide a generalizable proof of concept based on motion perception, we consider here the problem of judging the relative speed of moving dynamical textures and the impact of both average spatial frequency and average duration of temporal correlations.

## 3.1 Methods

The task was to discriminate the speed $v \in \mathbb{R}$ of MC stimuli moving with a horizontal central speed $\mathbf{v} = (v, 0)$. We assign as independent experimental variable the most represented spatial frequency $m_Z$, that we denote in the following $z$ for easier reading. The other parameters are set to the following values $\sigma_V = \frac{1}{t^\star z}$, $\theta_0 = \frac{\pi}{2}$, $\sigma_\Theta = \frac{\pi}{12}$, and $d_Z = 1.0$ c/°. Note that $\sigma_V$ is thus dependent of the value of $z$ (that is computed from $m_Z$ and $d_Z$, see Remark 2 and the supplementary ) to ensure that $t^\star = \frac{1}{\sigma_V z}$ stays constant. This parameter $t^\star$ controls the temporal frequency bandwidth, as illustrated on the middle of Figure 2. We used a two alternative forced choice (2AFC) paradigm. In each trial a grey fixation screen with a small dark fixation spot was followed by two stimulus intervals of 250 ms each, separated by a grey 250 ms inter-stimulus interval. The first stimulus had parameters $(v_1, z_1)$ and the second had parameters $(v_2, z_2)$. At the end of the trial, a grey screen appeared asking the participant to report which one of the two intervals was perceived as moving faster by pressing one of two buttons, that is whether $v_1 > v_2$ or $v_2 > v_1$.

Given reference values $(v^\star, z^\star)$, for each trial, $(v_1, z_1)$ and $(v_2, z_2)$ are selected so that

$$\begin{cases} v_i = v^\star, \ z_i \in z^\star + \Delta_Z \\ v_j \in v^\star + \Delta_V, \ z_j = z^\star \end{cases} \quad \text{where} \quad \begin{cases} \Delta_V = \{-2, -1, 0, 1, 2\}, \\ \Delta_Z = \{-0.48, -0.21, 0, 0.32, 0.85\}, \end{cases}$$

where $(i, j) = (1, 2)$ or $(i, j) = (2, 1)$ (i.e. the ordering is randomized across trials), and where $z$ values are expressed in cycles per degree (c/°) and $v$ values in °/s. Ten repetitions of each of the 25 possible combinations of these parameters are made per block of 250 trials and at least four such blocks were collected per condition tested. The outcome of these experiments are summarized by psychometric curves $\hat{\varphi}_{v^\star, z^\star}$, where for all $(v - v^\star, z - z^\star) \in \Delta_V \times \Delta_Z$, the value $\hat{\varphi}_{v^\star, z^\star}(v, z)$ is the empirical probability (each averaged over the typically 40 trials) that a stimulus generated with parameters $(v^\star, z)$ is moving faster than a stimulus with parameters $(v, z^\star)$.

To assess the validity of our model, we tested four different scenarios by considering all possible choices among $z^\star = 1.28$ c/°, $v^\star \in \{5°/\text{s}, 10°/\text{s}\}$, and $t^\star \in \{0.1s, 0.2s\}$, which corresponds to combinations of low/high speeds and a pair of temporal frequency parameters. Stimuli were generated on a Mac running OS 10.6.8 and displayed on a 20" Viewsonic p227f monitor with resolution $1024 \times 768$ at 100 Hz. Routines were written using Matlab 7.10.0 and Psychtoolbox 3.0.9 controlled the stimulus display. Observers sat 57 cm from the screen in a dark room. Three observers with normal or corrected to normal vision took part in these experiments. They gave their informed consent and the experiments received ethical approval from the Aix-Marseille Ethics Committee in accordance with the declaration of Helsinki.

## 3.2 Bayesian modeling

To make full use of our MC paradigm in analyzing the obtained results, we follow the methodology of the Bayesian observer used for instance in [13, 12, 8]. We assume the observer makes its decision using a Maximum A Posteriori (MAP) estimator $\hat{v}_z(m) = \operatorname*{argmin}_v [-\log(\mathbb{P}_{M|V,Z}(m|v, z)) - \log(\mathbb{P}_{V|Z}(v|z))]$ computed from some internal representation $m \in \mathbb{R}$ of the observed stimulus. For simplicity, we assume that the observer estimates $z$ from $m$ without bias. To simplify the numerical analysis, we assume that the likelihood is Gaussian, with a variance independent of $v$. Furthermore,

we assume that the prior is Laplacian as this gives a good description of the a priori statistics of speeds in natural images [2]:

$$\mathbb{P}_{M|V,Z}(m|v,z) = \frac{1}{\sqrt{2\pi}\sigma_z}e^{-\frac{|m-v|^2}{2\sigma_z^2}} \quad \text{and} \quad \mathbb{P}_{V|Z}(v|z) \propto e^{a_z v}1_{[0,v_{\max}]}(v). \tag{7}$$

where $v_{\max} > 0$ is a cutoff speed ensuring that $\mathbb{P}_{V|Z}$ is a well defined density even if $a_z > 0$. Both $a_z$ and $\sigma_z$ are unknown parameters of the model, and are obtained from the outcome of the experiments by a fitting process we now explain.

### 3.3 Likelihood and Prior Estimation

Following for instance [13, 12, 8], the theoretical psychophysical curve obtained by a Bayesian decision model is

$$\varphi_{v^\star,z^\star}(v,z) \stackrel{\text{def.}}{=} \mathbb{E}(\hat{v}_{z^\star}(M_{v,z^\star}) > \hat{v}_z(M_{v^\star,z}))$$

where $M_{v,z} \sim \mathcal{N}(v,\sigma_z^2)$ is a Gaussian variable having the distribution $\mathbb{P}_{M|V,Z}(\cdot|v,z)$.

The following proposition shows that in our special case of Gaussian prior and Laplacian likelihood, it can be computed in closed form. Its proof follows closely the derivation of [12, Appendix A], and can be found in the supplementary materials.

**Proposition 3.** *In the special case of the estimator* (3.2) *with a parameterization* (7)*, one has*

$$\varphi_{v^\star,z^\star}(v,z) = \psi\left(\frac{v - v^\star - a_{z^\star}\sigma_{z^\star}^2 + a_z\sigma_z^2}{\sqrt{\sigma_{z^\star}^2 + \sigma_z^2}}\right) \tag{8}$$

*where* $\psi(t) = \frac{1}{\sqrt{2\pi}}\int_{-\infty}^t e^{-s^2/2}\mathrm{d}s$ *is a sigmoid function.*

One can fit the experimental psychometric function to compute the perceptual bias term $\mu_{z,z^\star} \in \mathbb{R}$ and an uncertainty $\lambda_{z,z^\star}$ such that $\hat{\varphi}_{v^\star,z^\star}(v,z) \approx \psi\left(\frac{v-v^\star-\mu_{z,z^\star}}{\lambda_{z,z^\star}}\right)$.

*Remark* 5. Note that in practice we perform a fit in a log-speed domain *ie* we consider $\varphi_{\tilde{v}^\star,z^\star}(\tilde{v},z)$ where $\tilde{v} = \ln(1 + v/v_0)$ with $v_0 = 0.3°/$s following [13].

By comparing the theoretical and experimental psychopysical curves (8) and (3.3), one thus obtains the following expressions $\sigma_z^2 = \lambda_{z,z^\star}^2 - \frac{1}{2}\lambda_{z^\star,z^\star}^2$ and $a_z = a_{z^\star}\frac{\sigma_{z^\star}^2}{\sigma_z^2} - \frac{\mu_{z,z^\star}}{\sigma_z^2}$. The only remaining unknown is $a_{z^\star}$, that can be set as any negative number based on previous work on low speed priors or, alternatively estimated in future by performing a wiser fitting method.

### 3.4 Psychophysic Results

The main results are summarized in Figure 3 showing the parameters $\mu_{z,z^\star}$ in Figure 3(a) and the parameters $\sigma_z$ in Figure 3(b). Spatial frequency has a positive effect on perceived speed; speed is systematically perceived as faster as spatial frequency is increased, moreover this shift cannot simply be explained to be the result of an increase in the likelihood width (Figure 3(b)) at the tested spatial frequency, as previously observed for contrast changes [13, 12]. Therefore the positive effect could be explained by a negative effect in prior slopes $a_z$ as the spatial frequency increases. However, we do not have any explanation for the observed constant likelihood width as it is not consistent with the speed width of the stimuli $\sigma_V = \frac{1}{t^\star z}$ which is decreasing with spatial frequency.

### 3.5 Discussion

We exploited the principled and ecologically motivated parameterization of MC to ask about the effect of scene scaling on speed judgements. In the experimental task, MC stimuli, in which the spatial scale content was systematically varied (via frequency manipulations) around a central frequency of 1.28 c/° were found to be perceived as slightly faster at higher frequencies slightly slower at lower frequencies. The effects were most prominent at the faster speed tested, of 10 °/s relative to those at 5 °/s. The fitted psychometic functions were compared to those predicted by a Bayesian model in which the likelihood or the observer's sensory representation was characterised by a simple Gaussian. Indeed, for this small data set intended as a proof of concept, the model was able to explain

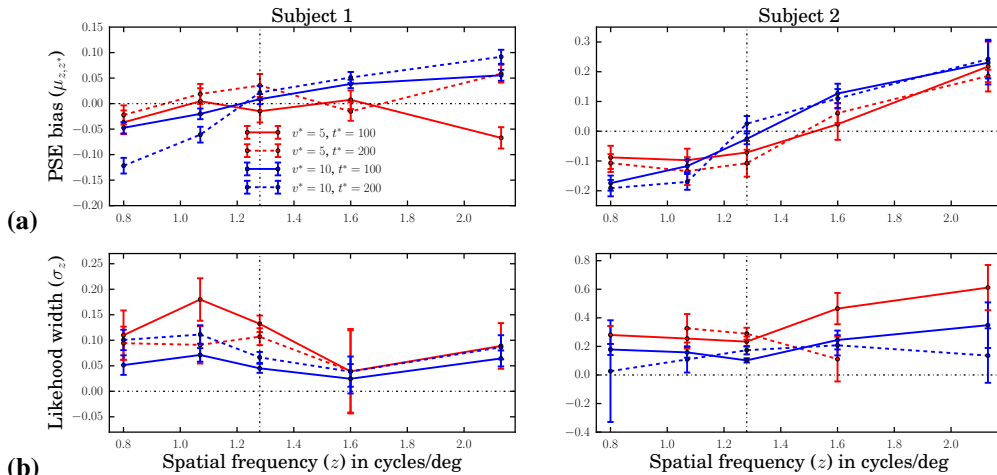

**Figure 3:** *2AFC speed discrimination results.* **(a)** Task generates psychometric functions which show shifts in the point of subjective equality for the range of test $z$. Stimuli of lower frequency with respect to the reference (intersection of dotted horizontal and vertical lines gives the reference stimulus) are perceived as going slower, those with greater mean frequency are perceived as going relatively faster. This effect is observed under all conditions but is stronger at the highest speed and for subject 1. **(b)** The estimated $\sigma_z$ appear noisy but roughly constant as a function of $z$ for each subject. Widths are generally higher for $v = 5$ (red) than $v = 10$ (blue) traces. The parameter $t^\star$ does not show a significant effect across the conditions tested.

these systematic biases for spatial frequency as shifts in our *a priori* on speed during the perceptual judgements as the likelihood width are constant across tested frequencies but lower at the higher of the tested speeds. Thus having a larger measured bias given the case of the smaller likelihood width (faster speed) is consistent with a key role for the prior in the observed perceptual bias.

A larger data set, including more standard spatial frequencies and the use of more observers, is needed to disambiguate the models predicted prior function.

## 4 Conclusions

We have proposed and detailed a generative model for the estimation of the motion of images based on a formalization of small perturbations from the observer's point of view during parameterized rotations, zooms and translations. We connected these transformations to descriptions of ecologically motivated movements of both observers and the dynamic world. The fast synthesis of naturalistic textures optimized to probe motion perception was then demonstrated, through fast GPU implementations applying auto-regression techniques with much potential for future experimentation. This extends previous work from [10] by providing an axiomatic formulation. Finally, we used the stimuli in a psychophysical task and showed that these textures allow one to further understand the processes underlying speed estimation. By linking them directly to the standard Bayesian formalism, we show that the sensory representations of the stimulus (the likelihoods) in such models can be described directly from the generative MC model. In our case we showed this through the influence of spatial frequency on speed estimation. We have thus provided just one example of how the optimized motion stimulus and accompanying theoretical work might serve to improve our understanding of inference behind perception. The code associated to this work is available at https://jonathanvacher.github.io.

## Acknowledgements

We thank Guillaume Masson for useful discussions during the development of the experiments. We also thank Manon Bouyé and Élise Amfreville for proofreading. LUP was supported by EC FP7-269921, "BrainScaleS". The work of JV and GP was supported by the European Research Council (ERC project SIGMA-Vision). AIM and LUP were supported by SPEED ANR-13-SHS2-0006.

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
