[Supplementary Material · MotionClouds-NIPS-supplementary.pdf]

# Biologically Inspired Dynamic Textures for Probing Motion Perception
## *Supplementary Material*

**Jonathan Vacher**
CNRS UNIC and Ceremade
Univ. Paris-Dauphine
75775 Paris Cedex 16, FRANCE
vacher@ceremade.dauphine.fr

**Andrew Isaac Meso**
Institut de Neurosciences de la Timone
UMR 7289 CNRS/Aix-Marseille Université
13385 Marseille Cedex 05, FRANCE
andrew.meso@univ-amu.fr

**Laurent Perrinet**
Institut de Neurosciences de la Timone
UMR 7289 CNRS/Aix-Marseille Université
13385 Marseille Cedex 05, FRANCE
laurent.perrinet@univ-amu.fr

**Gabriel Peyré**
CNRS and Ceremade
Univ. Paris-Dauphine
75775 Paris Cedex 16, FRANCE
peyre@ceremade.dauphine.fr

## Abstract

This document present additional results for the paper "Biologically Inspired Dynamic Textures for Probing Motion Perception". In particular it proves the equivalence between the initial (spatial or spectral) definition of MC and a sPDE formulation, which is useful both for fast numerical simulation and also for fast evaluation of the conditional density of MC (in order to use it as likelihood for psychophysical data analysis).

## 1 Graphical Display of MC

We recall that MC are stationary Gaussian random field with a parameterized power spectrum having the form

$$\forall (\xi, \tau) \in \mathbb{R}^3, \ \hat{\gamma}(\xi, \tau) = \frac{\mathbb{P}_Z (\|\xi\|)}{\|\xi\|^2} \mathbb{P}_\Theta (\angle\xi) \mathcal{L}(\mathbb{P}_{\|V - v_0\|}) \left( \|v_0\| \cos(\angle v_0 - \angle\xi) - \frac{\tau}{\|\xi\|} \right). \quad (1)$$

Similarly as was previously proposed in [4]. We show in Figure 1 two examples of such stimuli for different spatial frequency bandwidths. In particular, by tuning this bandwidth we could dissociate their respective role in action and perception [4, 5]. Extending the study of visual perception to other dimensions, such as orientation or speed bandwidths, should provide essential data to titrate their respective role in motion integration.

## 2 sPDE Formulation and Numerics

The formulation of the MC gives an explicit parameterization (1) of the covariance over the Fourier domain. We show here that it can be equivalently discretized by the solutions of a local PDE driven by a Gaussian noise. This formulation is important since we aim to deal with dynamic stimulation, which should be described by a causal equation which is local in time. This is indeed crucial to offer a fast simulation algorithm (see Section 2.5) and to offer a coherent Bayesian inference framework, as shown in Section 3.

A                                                    B

Figure 1: Broadband vs. narrowband stimuli. We show in (**A**) and (**B**) instances of the same Motion Clouds with different frequency bandwidths $\sigma_Z$, while all other parameters (such as $z_0$) are kept constant. The top column displays iso-surfaces of the spectral envelope by displaying enclosing volumes at different energy values with respect to the peak amplitude of the Fourier spectrum. The bottom column shows an isometric view of the faces of the movie cube. The first frame of the movie lies on the x-y plane, the x-t plane lies on the top face and motion direction is seen as diagonal lines on this face (vertical motion is similarly see in the y-t face). The Motion Cloud with the broadest bandwidth is thought to best represent natural stimuli, since, as those, it contains many frequency components. (**A**) $\sigma_Z = 0.25$, (**B**) $\sigma_Z = 0.0625$.

## 2.1 Dynamic Textures as Solutions of sPDE

A MC $I$ with speed $v_0$ can be obtained from a MC $I_0$ with zero speed by the constant speed time warping

$$I(x, t) \stackrel{\text{def.}}{=} I_0(x - v_0 t, t). \tag{2}$$

We now restrict our attention to $I_0$.

We consider Gaussian random fields defined by a stochastic partial differential equation (sPDE) of the form

$$\mathcal{D}(I_0) = \frac{\partial W}{\partial t}(x) \quad \text{where} \quad \mathcal{D}(I_0) \stackrel{\text{def.}}{=} \frac{\partial^2 I_0}{\partial t^2}(x) + \alpha \star \frac{\partial I_0}{\partial t}(x) + \beta \star I_0(x) \tag{3}$$

This equation should be satisfied for all $(x, t)$, and we look for Gaussian fields that are stationary solutions of this equation. In this sPDE, the driving noise $\frac{\partial W}{\partial t}$ is white in time (i.e. corresponding to the temporal derivative of a Brownian motion in time) and has a 2-D covariance $\Sigma_W$ in space and $\star$

is the spatial convolution operator. The parameters $(\alpha, \beta)$ are 2-D spatial filters that aim at enforcing an additional correlation in time of the model. Section 2.2 explains how to choose $(\alpha, \beta, \Sigma_W)$ so that the stationary solutions of (3) have the power spectrum given in (1) (in the case that $v_0 = 0$), i.e. are motion clouds.

This sPDE formulation is important since we aim to deal with dynamic stimulation, which should be described by a causal equation which is local in time. This is crucial for numerical simulation (as explained in Section 2.5) but also to simplify the application of MC inside a bayesian model of psychophysical experiments (see Section 3).

While it is beyond the scope of this paper to study theoretically this equation, one can show existence and uniqueness results of stationary solutions for this class of sPDE under stability conditions on the filers $(\alpha, \beta)$ (see for instance [8]) that we found numerically to be always satisfied in our simulations. Note also that one can show that in fact the stationary solutions to (3) all share the same law. These solutions can be obtained by solving the sODE (4) forward for time $t > t_0$ with arbitrary boundary conditions at time $t = t_0$, and letting $t_0 \to -\infty$. This is consistent with the numerical scheme detailed in Section 2.5.

## 2.2 Equivalence Between Spectral and sPDE MC Formulations

The sPDE equation (3) corresponds to a set of independent stochastic ODEs over the spatial Fourier domain, which reads, for each frequency $\xi$,

$$\forall t \in \mathbb{R}, \quad \frac{\partial^2 \hat{I}_0(\xi, t)}{\partial t^2} + \hat{\alpha}(\xi) \frac{\partial \hat{I}_0(\xi, t)}{\partial t} + \hat{\beta}(\xi) \hat{I}_0(\xi, t) = \hat{\sigma}_W(\xi) \hat{w}(\xi, t) \tag{4}$$

where $\hat{I}_0(\xi, t)$ denotes the Fourier transform with respect to the space variable $x$ only. Here, $\hat{\sigma}_W(\xi)^2$ is the spatial power spectrum of $\frac{\partial W}{\partial t}$, which means that

$$\Sigma_W(x, y) = c(x - y) \quad \text{where} \quad \hat{c}(\xi) = \hat{\sigma}_W^2(\xi). \tag{5}$$

Here $\hat{w}(\xi, t) \sim \mathcal{N}(0, 1)$ and $w$ is a white noise in space and time. This formulation makes explicit that $(\hat{\alpha}(\xi), \hat{\beta}(\xi))$ should be chosen in order to make the temporal covariance of the resulting process equal (or at least approximate) the temporal covariance appearing in (1) in the motion-less setting (since we deal here with $I_0$), i.e. when $v_0 = 0$. This covariance should be localized around 0 and non-oscillating. It thus makes sense to constrain $(\hat{\alpha}(\xi), \hat{\beta}(\xi))$ for the corresponding ODE (4) to be critically damped, which corresponds to imposing the following relationship

$$\forall \xi, \quad \hat{\alpha}(\xi) = \frac{2}{\hat{\nu}(\xi)} \quad \text{and} \quad \hat{\beta}(\xi) = \frac{1}{\hat{\nu}^2(\xi)}$$

for some relaxation step size $\hat{\nu}(\xi)$. The model is thus solely parameterized by the noise variance $\hat{\sigma_W}(\xi)$ and the characteristic time $\hat{\nu}(\xi)$.

The following proposition shows that the sPDE model (3) and the motion cloud model (1) are identical for an appropriate choice of function $\mathbb{P}_{\|V - v_0\|}$.

**Proposition 1.** *When considering*

$$\forall r > 0, \quad \mathbb{P}_{\|V - v_0\|}(r) = \mathcal{L}^{-1}(h)(r/\sigma_V) \quad \text{where} \quad h(u) = (1 + u^2)^{-2} \tag{6}$$

*where $\mathcal{L}$ is defined in* (1)*, equation* (4) *admits a solution $I$ which is a stationary Gaussian field with power spectrum* (1) *when setting*

$$\hat{\sigma}_W^2(\xi) = \frac{1}{\hat{\nu}(\xi) \|\xi\|^2} \mathbb{P}_Z(\|\xi\|) \mathbb{P}_\Theta(\angle \xi), \quad \text{and} \quad \hat{\nu}(\xi) = \frac{1}{\sigma_V \|\xi\|}. \tag{7}$$

*Proof.* For this proof, we denote $I^{\text{MC}}$ the motion cloud defined by (1), and $I$ a stationary solution of the sPDE defined by (3). We aim at showing that under the specification (7), they have the same covariance. This is equivalent to show that $I_0^{\text{MC}}(x, t) = I^{\text{MC}}(x + ct, t)$ has the same covariance as $I_0$. One shows that for any fixed $\xi$, equation (4) admits a unique (in law) stationary solution $\hat{I}_0(\xi, \cdot)$ which is a stationary Gaussian process of zero mean and with a covariance which is $\hat{\sigma}_W^2(\xi) r \star \bar{r}$ where $r$ is the impulse response (i.e. taking formally $a = \delta$) of the ODE $r'' + 2r'/u + r''/u^2 = a$

where we denoted $u = \hat{\nu}(\xi)$. This impulse response is easily shown to be $r(t) = te^{-t/u}\mathbb{1}_{\mathbb{R}+}(t)$. The covariance of $\hat{I}_0(\xi, \cdot)$ is thus, after some computation, equal to $\hat{\sigma}_W^2(\xi)r \star \bar{r} = \hat{\sigma}_W^2(\xi)h(\cdot/u)$ where $h(t) \propto (1 + |t|)e^{-|t|}$. Taking the Fourier transform of this equality, the power spectrum $\hat{\gamma}_0$ of $I_0$ thus reads

$$\hat{\gamma}_0(\xi, \tau) = \hat{\sigma}_W^2(\xi)\hat{\nu}(\xi)h(\hat{\nu}(\xi)\tau) \quad \text{where} \quad h(u) = \frac{1}{(1 + u^2)^2}$$

and where it should be noted that this $h$ function is the same as the one introduced in (6). The covariance $\gamma^{\text{MC}}$ of $I^{\text{MC}}$ and $\gamma_0^{\text{MC}}$ of $I_0^{\text{MC}}$ are related by the relation

$$\hat{\gamma}_0^{\text{MC}}(\xi, \tau) = \hat{\gamma}^{\text{MC}}(\xi, \tau - \langle \xi, v_0 \rangle) = \frac{1}{\|\xi\|^2}\mathbb{P}_Z(\|\xi\|)\mathbb{P}_\Theta(\angle\xi)\,h\left(-\frac{\tau}{\sigma_V\|\xi\|}\right).$$

where we used the expression (1) for $\hat{\gamma}^{\text{MC}}$ and the value of $\mathcal{L}(\mathbb{P}_{\|V-v_0\|})$ given by (6). Condition (7) guarantees that expression (2.2) and (2.2) coincide, and thus $\hat{\gamma}_0 = \hat{\gamma}_0^{\text{MC}}$. □

## 2.3 Expression for $\mathbb{P}_{\|V-v_0\|}$

Equation (6) states that in order to obtain a perfect equivalence between the MC defined by (1) and by (3), the function has $\mathcal{L}^{-1}(h)$ to be well-defined. It means we need to compute the inverse of the transform of the linear operator $\mathcal{L}$

$$\forall u \in \mathbb{R}, \quad \mathcal{L}(f)(u) = 2\int_0^{\pi/2} f(-u/\cos(\varphi))\mathrm{d}\varphi.$$

to the function $h$. The following proposition gives a closed-form expression for this function, and shows in particular that it is a function in $L^1(\mathbb{R})$, i.e. it has a finite integral, which can be normalized to unity to define a density distribution. Figure 2 shows a graphical display.

**Proposition 2.** *One has*

$$\mathcal{L}^{-1}(h)(u) = \frac{2 - u^2}{\pi(1 + u^2)^2} - \frac{u^2(u^2 + 4)(\log(u) - \log(\sqrt{u^2 + 1} + 1))}{\pi(u^2 + 1)^{5/2}}.$$

*In particular, one has*

$$\mathcal{L}^{-1}(h)(0) = \frac{2}{\pi} \quad \text{and} \quad \mathcal{L}^{-1}(h)(u) \sim \frac{1}{2\pi u^3} \quad \text{when} \quad u \to +\infty.$$

*Proof.* The variable substitution $x = \cos(\varphi)$ allows to rewrite (2.3) as

$$\forall u \in \mathbb{R}, \quad \mathcal{L}(h)(u) = 2\int_0^1 h\left(-\frac{u}{x}\right)\frac{x}{\sqrt{1 - x^2}}\frac{\mathrm{d}x}{x}.$$

In such a form, we recognize a Mellin convolution which could be inverted by the use of Mellin convolution table. □

## 2.4 Parametrization of $\mathbb{P}_Z$

**Parametrization by mode and standard deviation**    The log-normal distribution could be written

$$\mathbb{P}_Z(z) \propto \frac{z_0}{z}e^{-\frac{\ln\left(\frac{z}{z_0}\right)^2}{2\ln\left(1 + \sigma_Z^2\right)}}.$$

The parameters $(z_0, \sigma_Z)$ are convenient to write the distribution but they do not reflect remarkable values of a log-normal random variable. Instead, we prefer to fix directly the mode $m_Z = \text{argmax}_z \mathbb{P}_Z(z)$ and standard deviation $d_Z = \sqrt{\int_{\mathbb{R}_+} z^2\mathbb{P}_Z(z)\mathrm{d}z}$. These couples of variable are linked by the following equations,

$$m_Z = \frac{z_0}{1 + \sigma_Z^2} \quad \text{and} \quad d_Z = z_0\sigma_Z^2(1 + \sigma_Z^2).$$

Such formula could be inverted by finding the unique positive root of

$$P(x) = x^2(1 + x^2)^2 - \frac{d_Z}{m_Z}$$

because $P(\sigma_Z) = 0$ and finally set $z_0 = m_Z(1 + \sigma_Z^2)$.

Figure 2: Functions $h$ and $\mathcal{L}^{-1}(h)$.

**Parametrization by mode and octave bandwidth**   Another choice would be to parametrize $\mathbb{P}_Z$ by its mode $m_Z$ and octave bandwidth $B_Z$ which is defined by

$$B_Z = \frac{\ln\left(\frac{z_+}{z_-}\right)}{\ln(2)}$$

where $(z_-, z_+)$ are the half-power cutoff frequencies *ie* verifies $\mathbb{P}_Z(z_-) = \mathbb{P}_Z(z_+) = \frac{\mathbb{P}_Z(m_Z)}{2}$. This last condition comes down to study the roots of the following polynomial

$$Q(X) = X^2 + 2\ln(1 + \sigma_Z^2)X - 2\ln(2)\ln(1 + \sigma_Z^2) + \frac{1}{2}\ln(1 + \sigma_Z^2)^2$$

where $X = \ln\left(\frac{z}{z_0}\right)$. It follows that

$$B_Z = \sqrt{\frac{8\ln(1 + \sigma_Z^2)}{\ln(2)}}.$$

Conversely,

$$\sigma_Z = \sqrt{\exp\left(\frac{\ln(2)}{8}B_Z^2\right) - 1}.$$

## 2.5   AR(2) Discretization of the sPDE

Most previous works (such as [3] for static and [4, 5] for dynamic textures) have used global Fourier-based approach that makes use of the explicit power spectrum expression 1. The main drawbacks of such an approach are: (i) it introduces an artificial periodicity in time and thus can only be used to synthesize a finite number of frames; (ii) the discrete computational grid may introduce artifacts, in particular when one of the bandwidths is of the order of the discretization step; (iii) these frames must be synthesized at once, before the stimulation, which prevents real-time synthesis.

To address these issues, we follow the previous works of [2, 10] and make use of an auto-regressive (AR) discretization of the sPDE (3). In contrast with these previous works, we use a second order AR(2) regression (in place of a first order AR(1) model). Using higher order recursions is crucial to be consistent with the continuous formulation (3). Indeed, numerical simulations show that AR(1) iterations lead to unacceptable temporal artifacts: in particular, the time correlation of AR(1) random fields typically decays too fast in time.

The discretization computes a (possibly infinite) discrete set of 2-D frames $(I_0^{(\ell)})_{\ell \geqslant \ell_0}$ separated by a time step $\Delta$, and we approach at time $t = \ell\Delta$ the derivatives as

$$\frac{\partial I_0(\cdot, t)}{\partial t} \approx \Delta^{-1}(I_0^{(\ell)} - I_0^{(\ell-1)}) \quad \text{and} \quad \frac{\partial^2 I_0(\cdot, t)}{\partial t^2} \approx \Delta^{-2}(I_0^{(\ell+1)} + I_0^{(\ell-1)} - 2I_0^{(\ell)}),$$

which leads to the following explicit recursion

$$\forall \ell \geqslant \ell_0, \quad I_0^{(\ell+1)} = (2\delta - \Delta\alpha - \Delta^2\beta) \star I_0^{(\ell)} + (-\delta + \Delta\alpha) \star I_0^{(\ell-1)} + \Delta^2 W^{(\ell)}, \qquad (8)$$

where $\delta$ is the 2-D Dirac distribution and where $(W^{(\ell)})_\ell$ are i.i.d. 2-D Gaussian field with distribution $\mathcal{N}(0, \Sigma_W)$, and $(I_0^{(\ell_0-1)}, I_0^{(\ell_0-1)})$ can be arbitrary initialized.

One can show that when $\ell_0 \to -\infty$ (to allow for a long enough "warmup" phase to reach approximate time-stationarity) and $\Delta \to 0$, then $I_0^\Delta$ defined by interpolating $I_0^\Delta(\cdot, \Delta\ell) = I^{(\ell)}$ converges (in the sense of finite dimensional distributions) toward a solution $I_0$ of the sPDE (3). We refer to [9] for a similar result in the 1-D case (stochastic ODE). We implemented the recursion (8) by computing the 2-D convolutions with FFT's on a GPU, which allows us to generate high resolution videos in real time, without the need to explicitly store the synthesized video.

## 3   Experimental Likelihood vs. the MC Model

In our paper, we propose to directly fit the likelihood $\mathbb{P}_{M|V,Z}(m|v,z)$ from the experimental psychophysical curve. While this makes sense from a data-analysis point of view, this required strong modeling hypothesis, in particular, that the likelihood is Gaussian with a variance $\sigma_z^2$ independent of the parameter $v$ to be estimated by the observer.

In this section, we direct a likelihood model directly from the stimuli, by making another (of course questionnable) hypothesis, that the observer uses a standard motion estimation process, based on the motion energy concept [1], that we adapt here to the MC distribution. In this setting, this corresponds to using a MLE estimator, and making use of the sPDE formulation of MC.

### 3.1   MLE Speed Estimator

We first show how to compute this MLE estimator. To be able to achieve this, the following proposition derive the sPDE satisfied by a motion cloud with a non-zero speed.

**Proposition 3.** *A MC I with speed $v_0$ can be defined as a stationary solution of the sPDE*

$$\mathcal{D}(I) + \langle \mathcal{G}(I), v_0 \rangle + \langle \mathcal{H}(I)v_0, v_0 \rangle = \frac{\partial W}{\partial t} \qquad (9)$$

*where $\mathcal{D}$ is defined in* (3), *$\partial_x^2 I$ is the hessian of I (second order spatial derivative), where*

$$\mathcal{G}(I) \overset{\text{def.}}{=} \alpha \star \nabla_x I + 2\partial_t \nabla_x I \quad and \quad \mathcal{H}(I) \overset{\text{def.}}{=} (\partial_x^2 I)$$

*and $(\alpha, \beta, \Sigma_W)$ are defined in Proposition 1.*

*Proof.* This follows by derivating in time the warping equation (2), denoting $y \overset{\text{def.}}{=} x + v_0 t$

$$\partial_t I_0(x,t) = \partial_t I(y,t) + \langle \nabla I(y,t), v_0 \rangle,$$
$$\partial_t^2 I_0(x,t) = \partial_t^2 I(y,t) + 2\langle \partial_t \nabla I(y,t), v_0 \rangle + \langle \partial_x^2 I(y,t)v_0, v_0 \rangle$$

and plugging this into (3) after remarking that the distribution of $\frac{\partial W}{\partial t}(x,t)$ is the same as the distribution of $\frac{\partial W}{\partial t}(x - v_0 t, t)$. $\qquad\qquad\square$

Equation (9) is useful from a Bayesian modeling perspective, because, informally, it can be interpreted as the fact that the Gaussian distribution of MC as the following appealing form, for any function $\mathcal{I} : \mathbb{R}^2 \times \mathbb{R} \to \mathbb{R}$

$$\mathbb{P}_I(\mathcal{I}) = \frac{1}{Z_I} \exp(-\|\mathcal{D}(\mathcal{I}) + \langle \mathcal{G}(\mathcal{I}), v_0 \rangle + \langle \mathcal{H}(\mathcal{I})v_0, v_0 \rangle\|_{\Sigma_W^{-1}}^2)$$

where $Z_I$ is a normalization constant which is independent of $v_0$ and

$$\|\mathcal{I}\|_{\Sigma_W^{-1}}^2 \overset{\text{def.}}{=} \langle \mathcal{I}, \mathcal{I} \rangle_{\Sigma_W^{-1}} \quad and \quad \langle \mathcal{I}_1, \mathcal{I}_2 \rangle_{\Sigma_W^{-1}} \overset{\text{def.}}{=} \int \int \frac{\hat{\mathcal{I}}_1(\xi,t)\hat{\mathcal{I}}_2(\xi,t)^*}{\hat{\sigma}_W^2(\xi)} \mathrm{d}\xi \mathrm{d}t$$

where $\hat{\sigma}_W$ is defined in (5).

This convenient formulation allows to re-write the MLE estimator of the horizontal speed $v$ parameter of a MC as

$$\hat{v}^{\text{MLE}}(\mathcal{I}) \overset{\text{def.}}{=} \underset{v}{\arg\max} \; \mathbb{P}_I(\mathcal{I}) \quad \text{where} \quad v_0 = (v, 0) \in \mathbb{R}^2$$

used to analyse psychophysical experiments as

$$\hat{v}^{\text{MLE}}(\mathcal{I}) = \underset{v}{\arg\min} \; \|\mathcal{D}(\mathcal{I}) + v\langle\mathcal{G}(\mathcal{I}), (1,0)\rangle + v^2\langle\mathcal{H}(\mathcal{I})(1,0), (1,0)\rangle\|^2_{\Sigma_W^{-1}} \tag{10}$$

where we used the fact that the normalizing constant $Z_I$ is independent of $v_0$. Expanding the squares shows that (10) is the optimization of a fourth order polynomial, whose solution can be computed in closed form as one of the roots of the derivative of this polynomial, which is hence a third order polynomial.

## 3.2 MLE Modeling of the Likelihood

In our paper, following several previous works such as [7, 6], we assumed the existence of an internal representation parameter $m$, which was assumed to be a scalar, with a Gaussian distribution conditioned on $(v, z)$. We explore here the possibility that this internal representation could be directly obtained from the stimuli by the usage by the observer of an "optimal" speed detector (an MLE estimate).

Denoting $I_{v,z}$ a MC, which is a random Gaussian field of power spectrum (1), with central speeds $v_0 = (v, 0)$ and central spacial frequency $z$ (the other parameters being fixed as explained in the experimental section of the paper), this means that we consider the internal representation as being the following scalar random variable

$$M_{v,z} \overset{\text{def.}}{=} \hat{v}_z^{\text{MLE}}(I_{v,z}) \quad \text{where} \quad \hat{v}_z^{\text{MLE}}(\mathcal{I}) \overset{\text{def.}}{=} \underset{v}{\arg\max} \; \mathbb{P}_{M|V,Z}(\mathcal{I}|v, z), \tag{11}$$

As detailed in (10) it can be efficiently computed numerically.

As shown in Figure 3(a), we observed that $M_{v,z}$ is well approximated by a Gaussian random variable. Its mean is nearly constant and very close to $v$, and Figure 3(b) shows the evolution of its variance. Our main finding is that this optimal estimation model (using an MLE) is not consistent with the experimental finding because the estimated standard deviations of observers don't show a decreasing behavior as in Figure 3(b).

(a) Histogram

(b) Standard deviation

Figure 3: Estimates of $M_{v,z}$ defined by (11) and its standard deviation as a function of $z$.

## 3.3 Prior slope and Likelihood width fitting

In Section 3 we use equations

$$\sigma_z^2 = \lambda_{z,z^\star}^2 - \frac{1}{2}\lambda_{z^\star,z^\star}^2 \quad \text{and} \quad a_z = a_{z^\star}\frac{\sigma_{z^\star}^2}{\sigma_z^2} - \frac{\mu_{z,z^\star}}{\sigma_z^2}$$

to determine $a_z$ and $\sigma_z$. The slopes $a_z$ are noisy due to the quotient $\frac{\sigma_{z^\star}^2}{\sigma_z^2}$ therefore we only show some of the best fit in Figure 4 when the approximation $\sigma_z^2$ constant holds.

Figure 4: Example of decreasing $a_z$. The unknown $a_{z^\star}$ choosen so that $\sum_z a_z^2$ is minimum.

# 4 Proofs

## 4.1 Proof of Proposition 2

We recall the expression of the covariance

$$\forall (x,t) \in \mathbb{R}^3, \quad \gamma(x,t) = \int \int_{\mathbb{R}^2} c_g(\varphi_a(x-\nu t))\mathbb{P}_V(\nu)\mathbb{P}_A(a)\mathrm{d}\nu\mathrm{d}a \qquad (12)$$

We denote $(\theta,\varphi,z,r) \in \Gamma = [-\pi,\pi)^2 \times \mathbb{R}_+^2$ the set of parameters. According to Proposition 1, the covariance of $I$ is $\gamma$ defined by (12). Denoting $h(x,t) = c_g(zR_\theta(x-\nu t))$, one has, in the sense of distributions (taking the Fourier transform with respect to $(x,t)$)

$$\hat{h}(\xi,\tau) = z^{-2}\hat{g}(z^{-1}R_\theta(\xi))^2\delta_\mathcal{Q}(\nu) \quad \text{where} \quad \mathcal{Q} = \left\{\nu \in \mathbb{R}^2 \ ; \ \tau + \langle \xi, \nu \rangle = 0 \right\}.$$

Taking the Fourier transform of (12) and using this computation, one has

$$\hat{\gamma}(\xi,\tau) = \int_\Gamma \frac{1}{z^2}|\hat{g}\left(z^{-1}R_\theta(\xi)\right)|^2\delta_\mathcal{Q}(v_0 + r(\cos(\varphi),\sin(\varphi)))\mathbb{P}_\Theta(\theta)\mathbb{P}_Z(z)\mathbb{P}_{\|V-v_0\|}(r) \, \mathrm{d}\theta \, \mathrm{d}z \, \mathrm{d}r \, \mathrm{d}\varphi.$$

In the special case of $g$ being a grating, i.e. $|\hat{g}|^2 = \delta_{\xi_0}$, one has in the sense of distributions

$$z^{-2}|\hat{g}\left(z^{-1}R_\theta(\xi)\right)|^2 = \delta_\mathcal{B}(\theta,z) \quad \text{where} \quad \mathcal{B} = \left\{(\theta,z) \ ; \ z^{-1}R_\theta(\xi) = \xi_0\right\}.$$

Observing that $\delta_\mathcal{Q}(\nu)\delta_\mathcal{B}(\theta,z) = \delta_\mathcal{C}(\theta,z,r)$ where

$$\mathcal{C} = \left\{(\theta,z,r) \ ; \ z = \|\xi\|, \ \theta = \angle\xi, \ r = -\frac{\tau}{\|\xi\|\cos(\angle\xi - \varphi)} - \frac{\|v_0\|\cos(\angle\xi - \angle v_0)}{\cos(\angle\xi - \varphi)}\right\}$$

one obtains the desired formula.

## 4.2 Proof of Proposition 3

One has the closed form expression for the MAP estimator

$$\hat{v}_z(m) = m - a_z\sigma_z^2,$$

and hence, denoting $\mathcal{N}(\mu,\sigma^2)$ the Gaussian distribution of mean $\mu$ and variance $\sigma^2$,

$$\hat{v}_z(M_{v,z}) \sim \mathcal{N}(v - a_z\sigma_z^2, \sigma_z^2)$$

where $\sim$ means equality of distributions. One thus has

$$\hat{v}_{z^\star}(M_{v,z^\star}) - \hat{v}_z(M_{v^\star,z}) \sim \mathcal{N}(v - v^\star - a_{z^\star}\sigma_{z^\star}^2 + a_z\sigma_z^2, \sigma_{z^\star}^2 + \sigma_z^2),$$

which leads to the results by taking expectation.