[Reviews · NeurIPS 2015]

Submitted by Assigned_Reviewer_1

The paper describes a novel way to parameterize stimuli for generating space-time texture fields modeled after real-world deformations. The intent is to provide a more rigid framework by which scientists may psychophysically test motion perception. The model allows one to generate causal time-structured "Motion Clouds" in real-time. Motion Clouds are textured stimuli, which could be gabor-like or described by other functions. The model includes independent variables to describe motion, position, and warping. Warping describes texture zooming and rotations. They further suggest parameter distributions and stimulus properties that match well with existing psychophysical data. They use their model to generate stimuli for a speed discrimination psychophysical task and demonstrate their ability to assess how the stimulus spatial frequency affects speed discrimination. I interpret their contribution as a GPU ready method for generating real-time stimuli that is parameterized in a meaningful way for studying texture-motion related psychophysical phenomena.

A further advantage of this formalism is that it allows for statements of the model within a Bayesian framework. This sets the stage for psychophysical experiments. The main result of these (although the dataset is small) is that spatial frequency has a positive effect on perceived speed: stimuli of lower frequency with respect to the reference are perceived as going slower. The Bayesian model was able to explain these systematic biases for spatial frequency as shifts in priors on speed during the speed judgements. Overall I find the data part of this paper to be less satisfying because there is insufficient data for detailed analysis within the Bayesian frame work and the authors are unable to interpret some of their results.

Other comments:

The paper has some formatting problems (figure Legends not sufficiently different from text)

that make it difficult to read.

Note: The discussion section is cut off in the second sentence and doesn't continue. It seems as though they lost some text somewhere...

Summary: An interesting method for generating motion stimuli for psychophysical experiments.

Submitted by Assigned_Reviewer_2

The paper notes (line 269) that stationary dynamic Gaussian textures

can be produced using existing methods such as Fourier filtering of noise,

and says that the proposed method has the advantage of being able to produce stimuli in real-time.

However why is this necessary? Most psychological stimuli

are prepared in advance, one can imagine ways of easily handing any need

for on-the-fly generation.

As well, Gaussian textures are a very limited class of images.

Classic work by Julesz in the 1960s-70s explored the relation between texture discriminability and second-order statistics. The results seemed to indicate simply that the idea of nth-order statistics does not correspond in any simple way to human perception. But as well, Julesz was using the full second-order distribution, which provides much more modeling power than

is available with the Gaussian restriction used here.

Line 94, why is [3] nonparametric? Line 270, missing reference
Summary: This looks like careful and sophisticated work, however I am not able to understand the motivations and relevance for NIPS. It may be better suited for a forum on psychological methods.

Submitted by Assigned_Reviewer_3

Weak review.

Discussion section on page 7 appears to be cut-off mid-sentence. A good discussion section would be helpful for clarifying the importance and contributions of this work.

The impact of this work would be increased by providing the source code for the stimuli generation for use by other psychophysics researchers (as in the original motion cloud work).

p 6 line 308. "assess" is misspelled. p 8 line 421. "dynamic" is misspelled. p 8 line 428. "though" should be "through"
Summary: Paper presents a generative model of dynamic textures, shows that this model can be formulated as a stochastic PDE, and uses it in a psychophysics task to show textures with higher spatial frequencies are perceived to have higher speeds. The model and psychophysics result are important, however, the paper is difficult to follow in places and would benefit from careful editing.

Submitted by Assigned_Reviewer_4

This paper

provides a generative model for motion estimation based on small perturbations of observer's position by rotations, scaling and translation. Axiomatic definition of motion cloud stimuli from Ref. 8 was provide together with fast synthesis of naturalistic textures that can efficiently probe motion perception. The psychophysical results on judging the relative speed of moving

dynamical textures were inconclusive with the authors concluding that larger datasets are needed.

The paper appear to be hastily written with substantial number of typos (e.g. lines 101, 108, 270, 264, 366). In some cases the sentences were incomprehesible (e.g. lines 366-370).
Summary: This paper considers the generation of dynamic motion cloud stimuli. These stimuli were proposed before but here are derived in a more rigorous manner. Psychophysical results were not conclusive.

Submitted by Assigned_Reviewer_5

[Summary] The paper presents a a generative model for motion perception, which can be fitted into a Bayesian inference framework. This is derived from an axiomatic and biologically-driven model, and then is shown to be a generalization of the well-known luminance conservation equation.

[Originality and Significance] The contributions of this paper need to be confirmed by other reviewers with expertise.

[Clarity] The paper is mostly well written. But I have some difficulty in understanding some details, since I do not have a solid background in this area.

[Questions & comments] I did not notice major defect in the paper to my knowledge. But it would be more interesting if there could be any discussion of relationship between the proposed model and the dynamic textures by [3] (e.g, how the definition and/or properties are equivalent/relevant). This will provide a broader perspective to understand the framework.

@L94, it is said that "the most prominent method is the non-parametric Gaussian auto-regressive (AR) framework of [3]", the model presented in [3], however, is a parametric one (actually, a linear dynamic system). Is there any typo or misinterpretation here?

[Mesc.] Reference at L270 is not properly referred.
Summary: The paper presents a a biologically-inspired generative model for motion perception, which can be fitted into a Bayesian inference framework. I cannot confidently evaluate the significance here though I do not find any major defects.

Author Feedback
Author rebuttal: We are glad that all reviewers appreciated the significant technical contributions of our manuscript. There is a consensus in the range of comments that the manuscript should more clearly communicate its main contribution on the synthesis of textures optimized for psychophysics and based on a rigorous mathematical model of natural image transformations. In this response, we tackle those key points seeking to clearly demonstrate why we believe our manuscript makes an important contribution to NIPS. In addition, we will correct all errors and omissions pointed out in the original submission, some of which we acknowledge hindered the comprehension of the manuscript.

#Reviewer 1
Q: However I am not able to understand the motivations and relevance for NIPS.
A: We strongly believe that this contribution is directly relevant to NIPS. Indeed, this work is at the interface between mathematical modeling and psychophysics. NIPS has been a great avenue for previous similar work which includes several important works of Weiss and Simoncelli, two authors whose research groups have made seminal contributions to Bayesian modelling of perception.

Q: However why is this necessary? Most psychological stimuli are prepared in advance, one can imagine ways of easily handing any need for on-the-fly generation.
A: For experimental purposes, the number of generated video frames can be enormous. This framework is used in practice in different labs for psychophysics & neurophysiology and we found it to be essential for the acquisition of a substantial number of conditions and trials. In particular, there is a genuine difficulty when very large dynamic stimulus presentations are used in high resolution or over longer durations (stimulation for several minutes). We have faced these limitations in the past and so have many colleagues. The code, which will be openly available, will make the greatest contribution in these contexts

Q: As well, Gaussian textures are a very limited class of images. [...] The results seemed to indicate simply that the idea of nth-order statistics does not correspond in any simple way to human perception.
A: We agree that Gaussian textures are the simplest statistical models, and can only account for a limited range of perceptual phenomena. From a computer graphics perspective, these models have been shown to be surprisingly effective at capturing micro-scale details, and have recently been very popular at SIGGRAPH ("Gabor noise"). Moreover, these stimuli are designed to be used in perceptual experiments. Gaussian textures are "worst case scenario" textures, as they are composed of the densest mixture of random textons. Using sparser texture most often makes the job "easier" for the neural system; hence their usefulness to assess its efficiency e.g. for speed discrimination. Finally, related to the point above, our contribution is a first step toward more generic models, e.g. non-linear s-PDEs that include higher order correlations. We will add a short discussion about these issues in the conclusion, which was indeed lacking.

Q: Line 94, why is [3] nonparametric?
A: This is a typo, which will be corrected.

#Reviewer 2

Q: The paper appears to be hastily written with substantial number of typos [...]
A: There were indeed some errors in the previous submission. We will do a full proof checking for the final version.

#Reviewer 3

Q: [...] any discussion of relationship between the proposed model and the dynamic textures by [3]
A: This is indeed a good suggestion. [3] makes use of an AR-1 dynamical system, so it can only capture 1st order sPDE, while we make use of 2nd order PDE's. We will add this remark to the final version of the paper. We found this difference to be absolutely crucial for visual experiments to capture the correct correlation in time. A detailed discussion about AR textures can be found in [17].

#Reviewer 4

Q: The impact of this work would be increased by providing the source code for the stimuli generation for use by other psychophysics researchers.
A: We strongly agree. We will open-source the code and direct readers to the online github repository within the revised text.

#Reviewer 6

Q: I find the data part of this paper to be less satisfying because there is insufficient data...
A: The psychophysics part was meant as a proof of concept, to bridge the gap between propositions of the generative model and Bayesian modelling of speed perception. While more data would indeed extend the possible interpretations, we have importantly tested the formalised relationship between likelihoods and the motion cloud parameterization with our dataset by asking a novel question about frequency related adjustments to perceived speed.

#Reviewers 4 & 6

Q: Discussion section on page 7 appears to be cut-off mid-sentence.
A: In fact, a poor typesetting resulted in a part of the discussion to appear after Figure 3's caption. We apologise for the inconvenience.